# Investigation of the Time-Dependent Stability of a Coal Roadway under the Deep High-Stress Condition Based on the Cvisc Creep Model

Zhiliang Yang [1,*], Cun Zhang [2] and Donghui Yang [1]

1   School of Coal Engineering, Shanxi Datong University, Datong 037003, China; ydhname@163.com
2   School of Energy & Mining Engineering, China University of Mining & Technology (Beijing), Beijing 100083, China; cumt-zc@cumtb.edu.cn
*   Correspondence: 15993791548@163.com

**Abstract:** Creep is a fundamental property that naturally exists in some types of rock, which is significant for the long-term stability of roadways during the mining process. In this paper, the long-term strength of coal and rock were determined via laboratory experiments, and a Cvisc elasto-viscoplastic model was adopted and introduced in FLAC3D, based on the 31101 transport roadway in the Hongqinghe Coal Mine, to investigate the influence of creep on the stability of a deep high-stress roadway. The test results show that the long-term strength of 3-1 coal and sandy mudstone was 18.65 MPa and 39.95 MPa, respectively. The plastic zone, the deformation, and the damage of the roadway's surrounding rock displayed an obvious increase after being excavated for 720 d as the creep model was chosen. The plastic zone failure was modeled with shear-p (1090.7 $m^3$), shear-n (381.7 $m^3$), tension-n (98.4 $m^3$), and tension-p (30.8 $m^3$). The damage value had an obvious increment of 21.2% (0.053), and the deformation increased in the order of the two sidewalls (1978 mm), the roof (907 mm), and the floor (101 mm). The creep of the roadway can be divided into three stages: the accelerating stage, the decaying stage, and the stable stage. The creep speed of each stage is greatly affected by the presence or absence of anchor spray support: the creep speed of the bare roadway roof, sidewalls, and floor stability was 1.01, 1.02, and 0.12 mm/d, respectively. After anchor spray support, the creep velocity, correspondingly, decreased to 0.69, 0.37, and 0.12 mm/d, and the amount of surrounding rock damage decreased from 0.302 to 0.243. This indicates that the anchor spray support can significantly reduce the creep effect of the roadway. The Cvisc creep model was verified to be reliable and can provide guidance for deep high-stress coal roadway support.

**Keywords:** high-stress; coal roadway; creep; elasto-viscoplastic model; numerical simulation; time-dependent stability

## 1. Introduction

In recent years, with the increasing depletion of shallow coal resources in China and the increasing depth of coal mining, the difficulty in controlling the surrounding rock of deep roadways has significantly increased, and the problem of rock creep has become increasingly prominent. It has been verified that creep may cause a number of engineering disasters such as roof caving, floor heave, and side wall collapse, which increase the security risk of deep roadways. Therefore, considering the creep effect in the control of surrounding rock in deep roadways is of great significance for maintaining the long-term stability of roadways.

Creep is an important mechanical property of rocks and has been receiving much attention in recent decades. On one hand, physical testing is an effective means for studying numerous aspects of creep for rocks. In the literature, the effectiveness of the long-term strength of rocks has been determined via uniaxial compression experiments [1–4]. Several researchers have investigated the creep characteristics of rocks under various environmental

conditions, such as water, temperature, freeze–thaw, and acid–base solutions [5–12], and various stress paths such as direct tensile creep test, uniaxial step loading creep test, triaxial cycling creep test [13–15]. Rock creep behavior is monitored by acoustic emission (AE) [16,17], and the fracture morphology of rock creep failure is revealed by scanning electron microscopy (SEM) [18–20].

On the other hand, numerical simulation software can serve as a complementary means of investigation, such as FLAC3D software (5.0 version), RFPA [2D]software (Basic version), and Comsol software (6.0 version) [21–24]. In addition, various nonlinear models have been used to study creep, among which the important ones are the Burgers model [25,26], the viscoelastic plastic model [27–29], the creep damage model [30–35], and the creep hardening damage model [36,37]. The latter models are almost all based on the Burger model. Meanwhile, numerical simulation analysis considering rock creep factors is combined with on-site application in engineering practice. Kang et al. [38] studied the rock roadway creep for a kilometer-deep mine with UDEC numerical simulation. Huang et al. [39] explored the reason for the formation of asymmetric deformation in deep roadways over 1000 m and solved the long-term stability control. The mechanism of creep-induced coal burst has been put forward by researchers [40–43]. The simulation of long-term stability was carried out for gypsum pillars and coal pillars [44–47]. By means of the contrast analysis of creep numerical simulation and the physical simulation test, the mechanism of roadway floor heave has been studied [48]. Wei et al. [49] pointed out that an unstable rheological layer was the main cause of diverticulum damage creep. Dong et al. [50] constructed a coupled creep model of surrounding rock and rock bolt and proposed an improved support scheme. Under the conditions of a rock mass exposed to significant deformations of the rock layers, a special role is played by the mining support, which should be characterized by high strength but above all by flexibility to adapt to the moving rocks [51,52]. Zhang et al. [53] and Han et al. [54] respectively studied the effects of water and mining speed on the creep of distant rocks.

However, most of the above studies based on the Burger creep model study the creep of roadways, while the Burger model does not consider the plastic flow of deep rock masses. In addition, few studies delineate the creep of deep high-stress coal roadways. Therefore, it is necessary to study the creep of deep high-stress coal roadways based on the Cvisc model considering the plastic flow. In the present paper, the 31101 transportation roadway of the Hongqinghe Coal Mine is taken as the engineering background. The long-term strength of coal rock is estimated using the uniaxial compression test in the laboratory. The Cvisc creep model in FLAC3D is compared with the Mohr–Coulomb model, studying the evolution of plastic zone, damage, and displacement field in the case of deep high-stress coal roadways with or without creep participation. The objectives of this paper are as follows: (i) to determine the long-term strength of coal and rock; (ii) to compare the Cvisc creep model considering time effect and the Mohr–Coulomb model in numerical simulation; (iii) to verify whether the simulation results based on the Cvisc model are in good agreement with the field measurements.

## 2. Analysis of Engineering Situations

### 2.1. Engineering Geology

The Hongqinghe Coal Mine is located in Inner Mongolia, China, as shown in Figure 1. This mine produces approximately 15 million tons of coal annually, and the area of the mine field is 140.76 km$^2$. Its structural form is consistent with the regional coal-bearing strata. It is generally a westward-dipping monoclinic structure, and the coal-bearing strata belong to the typical Yan'an Formation of the Middle and Lower Jurassic. The 31101 mining panel is located southwest of the minefield. The strike length and the dip width of the panel are 1387 m and 235 m, respectively, with a 3° inclination and a 7.0 m thickness. The full coal seam, comprehensive mechanized mining method is adopted in this mining panel, and the full caving method is adopted for the post-mining space. For the first mining panel, the mean thickness of the No. 3-1 coal seam is 7.0 m, with an average

overburden depth of 720 m. The layout plan of the roadway driven along the goaf is shown in Figure 2. According to the geological survey, the immediate roof is composed of thick, gray, layered sandy mudstone and siltstone with a thickness of 5.20 m. The immediate floor is composed of grayish-black sandy mudstone of shale structure, with a thickness of 8.58 m. The comprehensive geological map of the mine is shown in Figure 3.

**Figure 1.** Location of the coal mine.

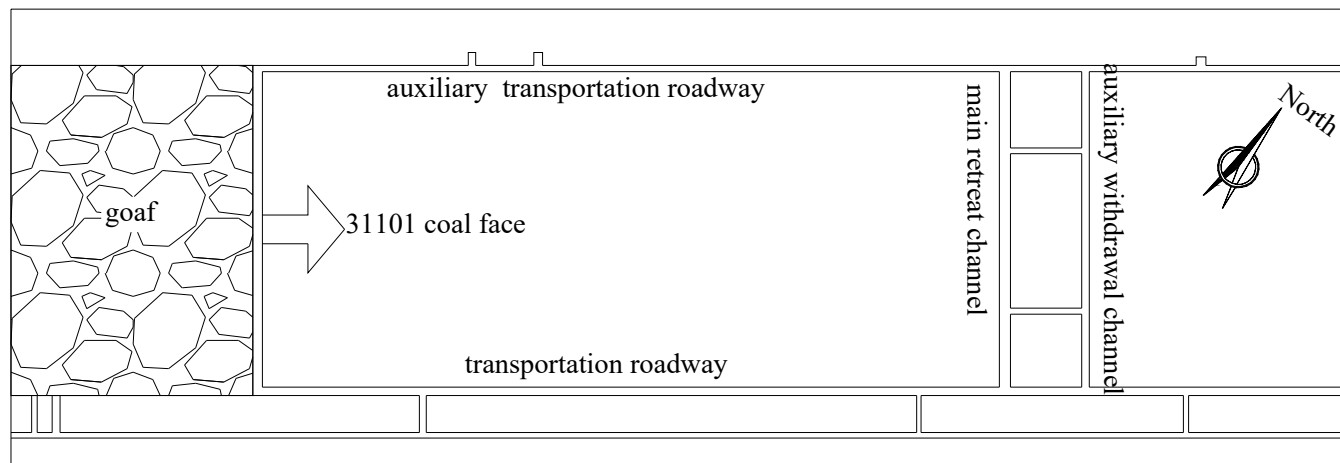

**Figure 2.** Mining panel and roadway layout.

| Thickness /m | Lithological columnar | Lithology | Lithology description |
|---|---|---|---|
| 41.29 | | Fine sandstone | Grayish white, massive, fine-grained arkose, medium sorting, sub angular, porous sandy argillaceous cementation, cross bedding. |
| 9.80 | | Middle conglomerate | Variegated, gravelly structure, blocky structure, mainly composed of quartz, feldspar, and debris, with poor sorting ability, sandy and muddy cementation, with a gravel diameter of 1-7cm. |
| 0.29 | | Fine conglomerate | Grey white, medium thick layered, poor sorting, good roundness, porous sand and mud cementation, with a gravel diameter of 0.02-0.04cm. |
| 7.80 | | Fine sandstone | Grey white, fine-grained structure, layered structure, containing dark minerals and mica, calcium mud cementation, cross bedding, and carbon debris. |
| 7.84 | | Siltstone | Light gray, silty structure, thick layered, locally mixed with coal chips, containing a small amount of mica fragments, horizontal bedding. |
| 5.20 | | Sandy mudstone | Gray, sandy and muddy texture, layered structure, containing a large amount of plant fossils, locally high carbon content, with horizontal and wavy bedding visible. |
| 6.36 | | 3-1 Coal | Black, asphalt glass luster, banded structure, layered structure, mainly composed of bright coal, followed by dark coal and vitrinite. |
| 8.58 | | Sandy mudstone | Light gray, blocky, with horizontal bedding and gently undulating bedding development, with a relatively flat cross-section. fossils of plant roots and stems can be seen at stepped fractures. |
| 18.20 | | Siltstone | Grayish white, partially grayish black, and mainly composed of quartz, feldspar, and debris, with argillaceous cementation. |

**Figure 3.** Stratigraphic column and geological description.

Both the roof and floor are weak and semi-hard rock layers that have poor stability. The mechanical properties of the rock mass degrade severely under high in situ stress; the deformation of rock masses becomes more and more serious, and the surrounding rock masses exhibit rheological deformation, as well as large and long-duration deformation. Especially, the 31101 transport roadway is excavated along the floor and shows obvious deformation as time passes. Along with the squeezing failure at the lower rib of the roadway, the floor heave is also very serious, as shown in Figure 4.

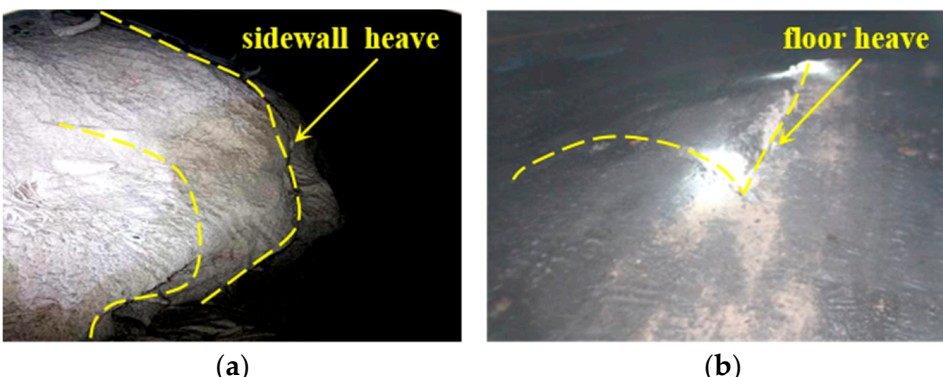

      (a)              (b)

**Figure 4.** Deformation plot of slot 31101. (**a**) Photo of sidewall failure profile. (**b**) Photo of floor failure profile.

*2.2. Support Parameters and Solutions*

The 31101 transport is 4900 m long, with a section size of 5600 mm × 4000 mm. Combined with the characteristics of high crustal stress, the surrounding rock is easily softened and expanded when encountering air weathering, water, and construction speed. The rock bolts are fixed to the surrounding rock along the entire length and the preliminary support parameters are as follows:

1.  The roof support adopts six left-hand-threaded steel longitudinal bolts; the specification is $\varphi$ 22 mm × $L$ 2400 mm; and the row and column spacing is 1000 mm × 1000 mm, adopting an arch-shaped high-strength tray, with a steel grade of no less than Q235 and a specification of 150 mm × 150 mm × 10 mm, with an arch height of not less than 34 mm, equipped with self-aligning ball pads and drag-reducing nylon washers.
2.  The sidewall support adopts five bolts, and the row and column spacing is 850 mm × 1000 mm, and its parameters are the same as for the roof. The sidewall has a 16# wire braided metal diamond mesh with a size of 5800 mm × 1100 mm.
3.  The floor is supported by two bolts, and each bolt is fitted with two resin cartridges MSSK2350 and MSK2350. The pretension force is not less than 300 N·m.

The different supporting schemes investigated in this study were as follows: (1) bolt perpendicular to the roadway roof and sidewall, (2) side bolt at an included angle of 45° relative to the roadway roof, sidewalls, and floor. After the anchoring was completed, full-section shotcrete was sprayed with C20 concrete to a thickness of 150 mm.

**3. Laboratory Experiments**

*3.1. Determination of Long-Term Strength of Coal and Rock*

The long-term strength of rock and coal that characterizes the aging condition is an important mechanical index to evaluate the stability of rock mass engineering. At present, there are two prevalent methods to calculate the long-term strength of coal and rock [55]:

1.  By creep test. In this method, the long-term strength of rock is determined by creep experiments, which are roughly equivalent to the softening critical load.
2.  By empirical formula. The calculation formula reads

$$\sigma_{cs} = K \cdot R_c \tag{1}$$

where $R_c$—rock uniaxial compressive strength, MPa; $K$—experience coefficient: expansive soft rock, $K = 0.3\sim0.5$; high-stress soft rock, $K = 0.5\sim0.7$; jointed soft rock, $K = 0.4\sim0.8$.

This paper takes the second approach after carrying out a series of uniaxial compression experiments in the laboratory. The whole experiment process adopts displacement control and loads the specimen at a constant loading rate of 0.1 mm/min, as shown in Figure 5. The experimental results (in Table 1) indicated that the compressive strengths of coal (No. 3-1) and sandy mudstone were 37.30 and 79.90 MPa, respectively. Based on Equation (1), we obtained the long-term strengths of 3-1 coal and sandy mudstone as 18.65 and 39.95 MPa, respectively.

**Table 1.** Values of uniaxial compressive strength (UCS) and modulus of elasticity (E) for rock and coal specimens.

| Specimen (No.) | UCS (Mpa) | E (Gpa) | Density (g·cm$^{-3}$) |
|:---:|:---:|:---:|:---:|
| A1 | 32.19 | 2.02 | 1.31 |
| A2 | 33.73 | 2.01 | 1.36 |
| A3 | 46.09 | 1.87 | 1.14 |
| B1 | 82.08 | 10.52 | 2.17 |
| B2 | 76.49 | 10.54 | 2.14 |
| B3 | 81.23 | 9.75 | 2.14 |

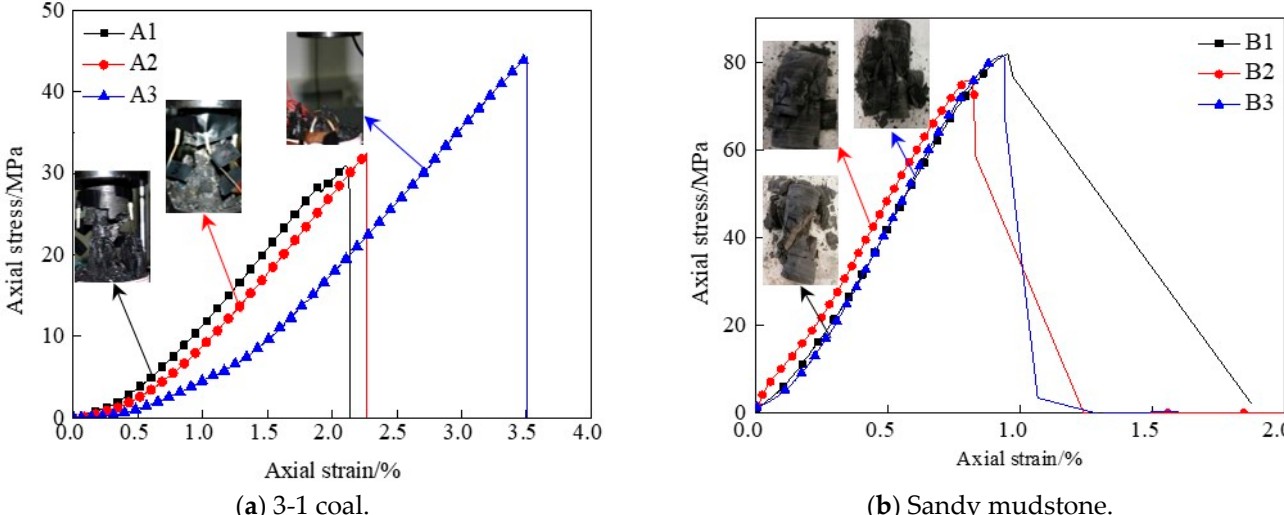

**Figure 5.** Stress–strain curve.

*3.2. Mineral Composition and Microstructure of Deep Surrounding Rocks*

The roof and floor are mainly composed of sandy mudstone, and the 31101 transportation roadway is excavated along the floor. The sandy mudstone is gray, with micrometer-scale wavy bedding and mudstone bands, containing plant leaf fossils. According to X-ray diffraction (XRD) and scanning electron microscopy (SEM), the main composition of the sandy mudstone is quartz 50.7%, potassium feldspar 3.6%, plagioclase 15.2%, clay mineral 30.5% (Yimeng mixed bed 6.4%, illite 5.9%, kaolinite 12.2%, chlorite 6%). The sandy mudstone is rich in clay (mainly illite) that shows strong swelling behavior, easily disintegrates, and softens when met with water, and the creep phenomenon of the surrounding rock is obvious during the excavation of water-rich sandy mudstone tunnels. The fracture morphology is magnified by 5000 times, and the surface of the fracture appears as a cluster of ductile pits, indicating that the sandy mudstone exhibits typical shear failure.

## 4. Cvisc Creep Numerical Simulation

*4.1. Elasto-Viscoplastic Creep Constitutive Model*

The Cvisc model introduces a Mohr–Coulomb plastic element based on the Burgers model, which assembles a series of Maxwell and Kelvin models [56]. In the Cvisc model, when the applied stress is smaller than the yield stress $\sigma_s$ of the Mohr–Coulomb criterion, the Cvisc model turns out to be the Burger model, and the rheological equation reads as Equation (2). Meanwhile, when the applied stress is larger than or equal to the yield stress $\sigma_s$, the element undergoes plastic flow based on the Mohr–Coulomb criterion, and the rheological equation is shown as Equation (3).

$$\varepsilon = \sigma \left[ \frac{1}{E_M} + \frac{t}{\eta_M} + \frac{1}{E_K} \left( 1 - \exp\left( -\frac{E_K}{\eta_K} \right) \right) \right] \quad \sigma < \sigma_s \tag{2}$$

$$\varepsilon = \sigma \left[ \frac{1}{E_M} + \frac{t}{\eta_M} + \frac{1}{E_K} \left( 1 - \exp\left( -\frac{E_K}{\eta_K} \right) \right) \right] + \varepsilon_p \quad \sigma \geq \sigma_s \tag{3}$$

where $\eta_M$—Maxwell viscosity coefficient, $\eta_K$—Kelvin viscosity coefficient, $E_M$—Maxwell modulus, $E_K$—Kelvin modulus, $\sigma_s$—yield stress, $\varepsilon_P$—plastic strain.

As the stress induced by coal mining is large in the deep layers, the Cvisc model can well describe the creep characteristics and plastic flow of coal/rock mass, as shown in Figure 6.

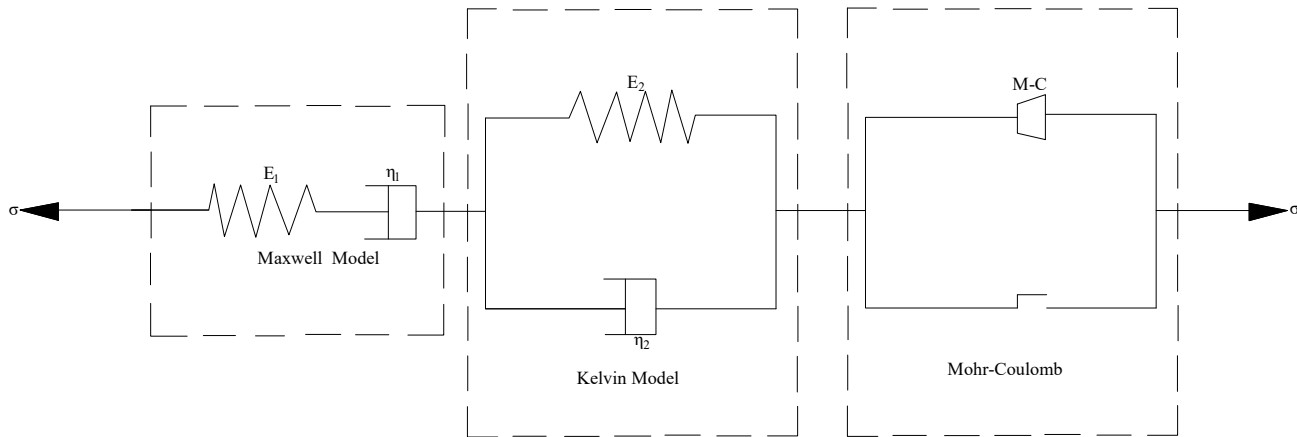

**Figure 6.** Cvisc model.

### *4.2. Numerical Model Description*

Based on the geological conditions of the surrounding rock for the 31101 transportation roadway, the numerical model was established using FLAC5.0, and the mesh was divided by local encryption and gradation. The numerical model, with the calculation range of 50 m(X) × 20 m(Y) × 30 m(Z), was divided into 38,000 zones and 42,672 grid points, and the shape of the 31101 transportation roadway is a rectangle, with width and height dimensions of 5.5 m × 4.0 m. The lateral and bottom boundary displacements are constrained, and the upper boundary represents the free surface. According to the hydraulic fracturing logging report of the Majihe Minefield where the Hongqinghe Mine is located, the In situ stress test data of 718 m and 731 m were selected in Table 2. The stress boundaries of the model were set based on the results of field in situ stress measurements: 19.0 Mpa in the z-direction, 21.0 Mpa in the y-direction, and 24.5 Mpa in the x-direction. The constitutive models of Mohr–Coulomb and Cvisc creep were used, respectively. The rheological parameters listed in Table 3 were comprehensively analyzed according to the literature [57–59] and engineering geological conditions. Figure 7b,c show the two different supporting schemes introduced in Section 2.2, and four monitoring points (A, B, C, and D) were preset at the centers of the two sidewalls, the roof, and the floor to evaluate the long-term change of the roadway. Combined with the support design of Hongqinghe 31101, the thickness of the shotcrete with C20 was 150 mm, and the main parameters and support density of the bolts are shown in Tables 4 and 5.

**Table 2.** The in situ stress test data.

| Buried Depth (m) | Maximum Principal Stress (MPa) | Intermediate Principal Stress (Mpa) | Minimum Principal Stress (Mpa) | Burst Orientation (°) |
|---|---|---|---|---|
| 718 | 24.21 | 21.40 | 16.51 | 45 |
| 731 | 21.00 | 20.65 | 16.81 | 45 |
| Average | 22.60 | 21.00 | 16.66 | 45 |

**Table 3.** The physical and mechanical parameters used for simulation.

| Rocktpe | Bulk (Gpa) | $G_K$ (Gpa) | $G_M$ (Gpa) | $\eta_K$ (Gpa.h) | $\eta_M$ (Gpa.h) | Cohesion (Mpa) | Internal Friction Angle (°) | Tensile Strength (Mpa) | Dilatancy Angle (°) |
|---|---|---|---|---|---|---|---|---|---|
| Siltstone | 2.02 | 132.6 | 6.1 | 937 | 23,079 | 1.60 | 26.1 | 1.75 | 11.6 |
| Sandy mudstone | 2.80 | 156.3 | 5.3 | 937 | 15,702 | 1.67 | 28.2 | 0.32 | 10.7 |
| 3-1 coal | 1.86 | 55.6 | 3.7 | 631 | 8703 | 1.28 | 21.8 | 0.15 | 15.3 |
| Sandy mudstone | 2.79 | 156.3 | 5.3 | 937 | 15,702 | 1.61 | 27.7 | 0.30 | 10.7 |
| Siltstone | 2.02 | 132.6 | 6.1 | 1331 | 23,079 | 1.60 | 26.1 | 1.75 | 11.6 |

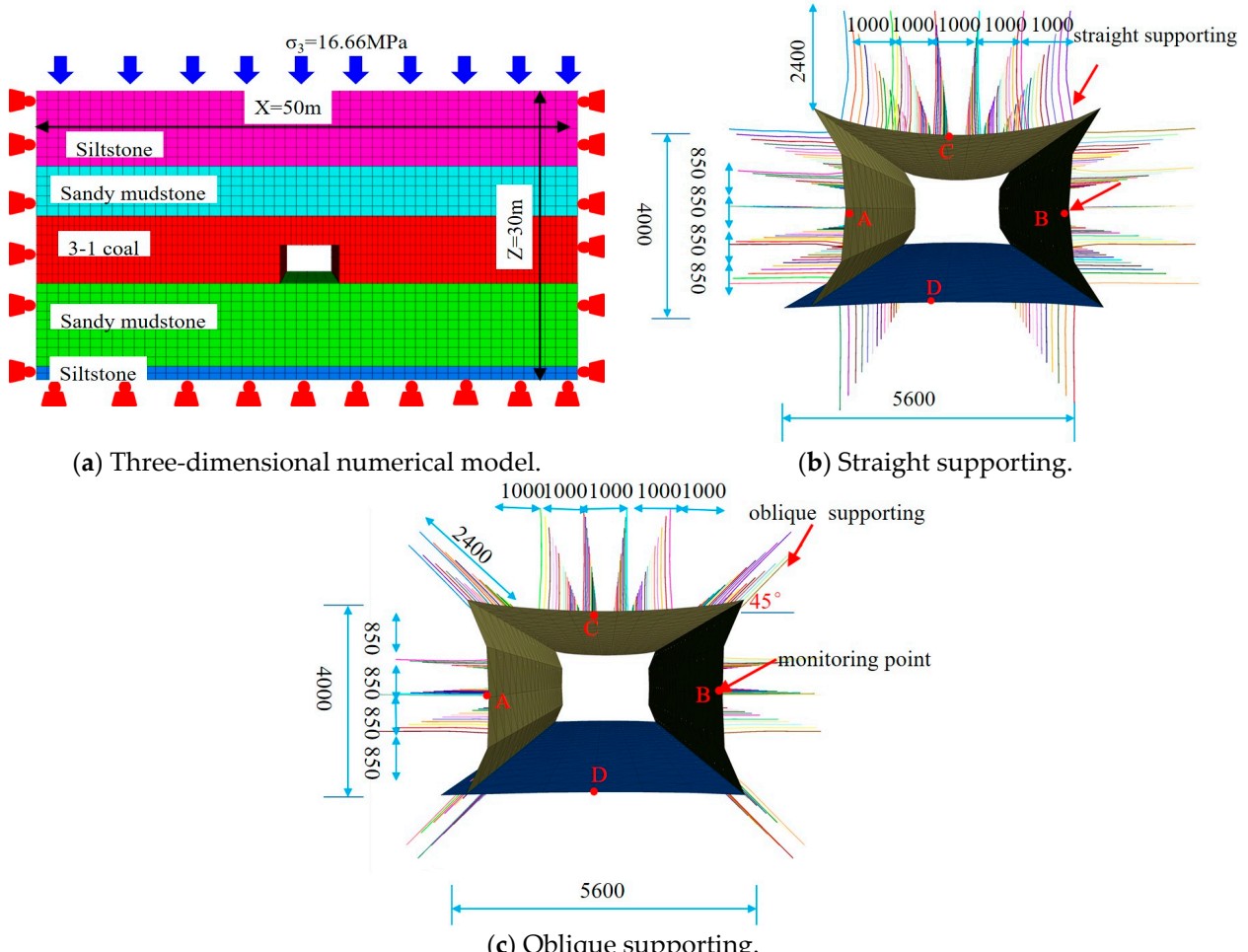

(**a**) Three-dimensional numerical model.  (**b**) Straight supporting.

(**c**) Oblique supporting.

**Figure 7.** Numerical model and supporting diagram.

**Table 4.** Shotcrete support parameters.

| Density (kg m$^{-3}$) | Elastic Modulus/Gpa | Poisson's Ratio | Internal Friction Angle/(°) | Cohesion (Mpa) | Tensile Strength (Mpa) |
|---|---|---|---|---|---|
| 2350 | 25.0 | 0.18 | 35 | 7.5 | 4.0 |

**Table 5.** The parameters of the rock bolt support adopted in numerical modeling.

| Rock Bolt | Length/m | E/Mpa | Tensile Strength/Mpa | Xcarea/m$^2$ | Unit Length Cement Cohesion/N | Unit Length Cement Stiffness N/m | Anchor Solid Perimeter/m |
|---|---|---|---|---|---|---|---|
| Free section | 1 | $200 \times 10^3$ | 250 | 0.00038 | 1 | 1 | 0.0785 |
| Anchor section | 1.2 | $200 \times 10^3$ | 250 | 0.00038 | $1 \times 10^6$ | $1.75 \times 10^7$ | 0.0785 |

### 4.3. Analysis of Plastic Zone

After the operation was completed, the FISH language was imported into FLAC to calculate the volume of the plastic zone. Figures 8 and 9 show the current shear damage stations and tensile damage state stations (by shear-n and tension-n), respectively. Figure 10 shows the previous shear damage station and tensile damage state stations (by shear-p and tension-p), respectively.

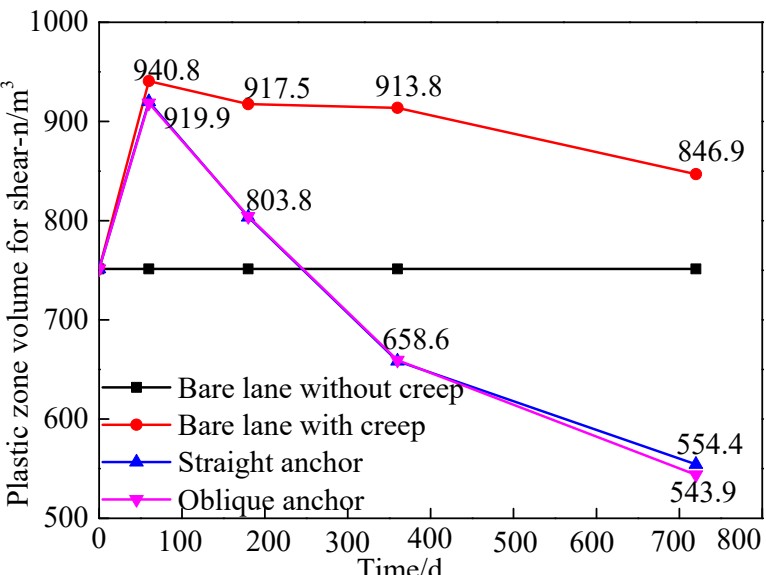

**Figure 8.** The volume variation of shear-n zones with time.

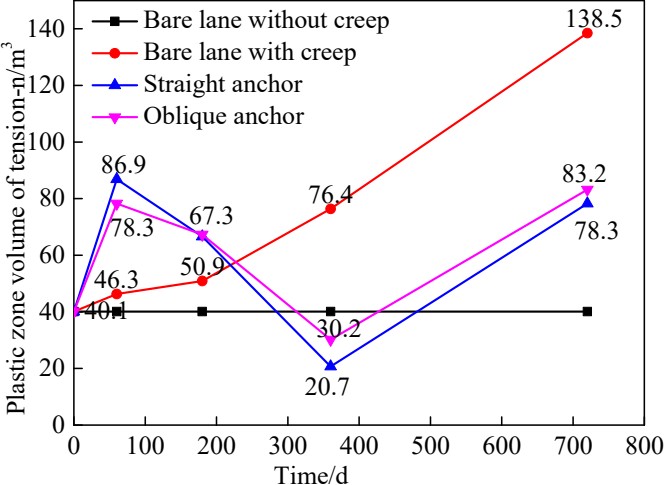

**Figure 9.** The volume variation of tension-n zones with time.

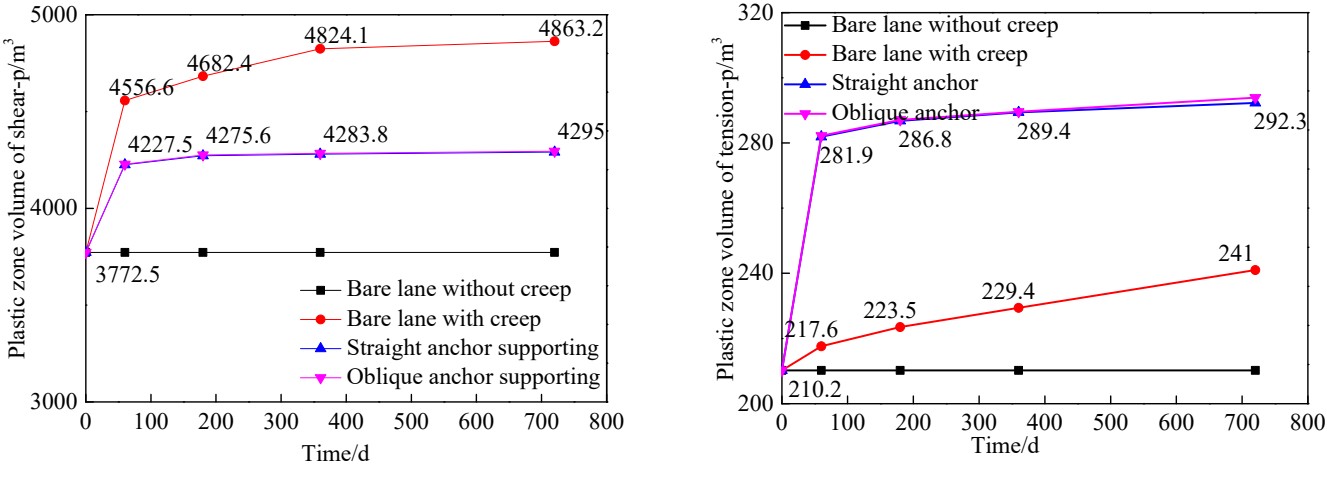

(**a**) The volume zone of shear-p.    (**b**) The volume zone of tension-p.

**Figure 10.** The volume variation of tension-p and shear-p zone with time.

### 4.3.1. The Volume Variation of Shear-n Zone

As shown in Figure 8, when the roadway had no creep and support, the shear-n zone volume was 751.5 m$^3$ and no longer changed with time.

When the roadway was subjected to creep, the zone volume of shear-n increased at 0~60 d, then slightly reduced at 60~360 d, and reduced at 360~720 d. The volumes of the shear-n zones for the bare roadway, the oblique anchor support, and the straight support were 846.9 m$^3$, 554.3 m$^3$, and 543.9 m$^3$, respectively.

### 4.3.2. The Volume Variation of Tension-n Zones

As shown in Figure 9, under the condition of no creep, the maximum volume of the tension-n zones was 40.1 m$^3$, and it no longer changed with time. While the creep deformation was considered for the bare lane, the volume of the tension-n zones sharply increased to 46.3 m$^3$ at 0~60 d, the increment in the tension-n zones was minimal at 60~180 d, while it was more obvious at 180~720 d (increase from 50.9 to 138.5 m$^3$). For the lane with creep deformation and supporting, the volume of the tension zones sharply increased to 86.9 m$^3$ for straight anchor supporting and the volume was 78.3 m$^3$ for oblique anchor supporting between 0 and 60 d; the tension-n volume slowly reduced at 60~180 d and sharply decayed at 180~360 d; then, it increased to 83.2 m$^3$ for the oblique anchor supporting and 78.3 m$^3$ for the straight anchor supporting at 720 d.

### 4.3.3. The Volume Variation of Tension-p and Shear-p Zones

As shown in Figure 10, for the bare lane with no creep deformation, the maximum volume of the shear-p and tension-p zones, respectively, was 3772.5 m$^3$ and 210.2 m$^3$, and it no longer changed with time. For the bare lane with creep deformation, these two variables rapidly increased at 0~60 d and steadily increased at 60~360 d; then, they slightly increased at 360~720 d. The volume of the shear-p and tension-p zones reached 4863.2 m$^3$ and 210.2 m$^3$, respectively. For the lane with creep deformation and supporting, the volume of the shear-p and tension-p zones, respectively, was 4295.2 m$^3$ and 292.3 m$^3$.

Compared to no creep, the increase in the volume of the different types of the plastic zone after 720 d of creep in the bare lane was, from largest to smallest, as follows: shear-p (1090.7 m$^3$) > shear-n (381.7 m$^3$) > tension-n (98.4 m$^3$) > tension-p (30.8 m$^3$). Compared with the creep for the bare lane, after bolting and shotcrete, the tension-p volume increased by 51.3 m$^3$, and the volume of other types of plastic zones decreased. When the straight anchor support was used, the reduction was, from large to small, shear-n (1170 m$^3$) > shear-p (568 m$^3$) > tension-n (60.2 m$^3$), and when the oblique anchor spray support was used, the reduction was, from large to small, shear-n (1212.1 m$^3$) > shear-p (568 m$^3$) > tension-n (55.3 m$^3$).

As shown in Figure 11, the failure of the surrounding rock began with the shear failure in the roof; then, tensile failure appeared in the sidewalls, and shear failure appeared in the floor. As the roof crack fissure developed, the failure became tensile failure.

The zones of tensile failure, shear failure, and the undestroyed zone developed from the inside to the outside of the surrounding rock. With increasing creep deformation, the area of tensile failure in the roof gradually increased, and the external shear failure range gradually expanded. Then, after bolting and shotcrete, the expansion trend of the plastic zone of the surrounding rock was suppressed, and the shear failure zone in the roof was effectively reduced. After roadway excavation (60~180 d), supporting and the surrounding rock work were implemented together. At the same time, the two sidewalls began to gradually transfer from the tension zone to the shearing zone. The bearing capacity was maintained and made the roadway easy to maintain. The shear plastic zone in the roof was further reduced 180~720 d after roadway excavation. Compared with the straight anchor, the corner anchor support can effectively inhibit the development of the shear plastic zone of the roof, and the straight anchor can restrain the expansion of the stretch zone.

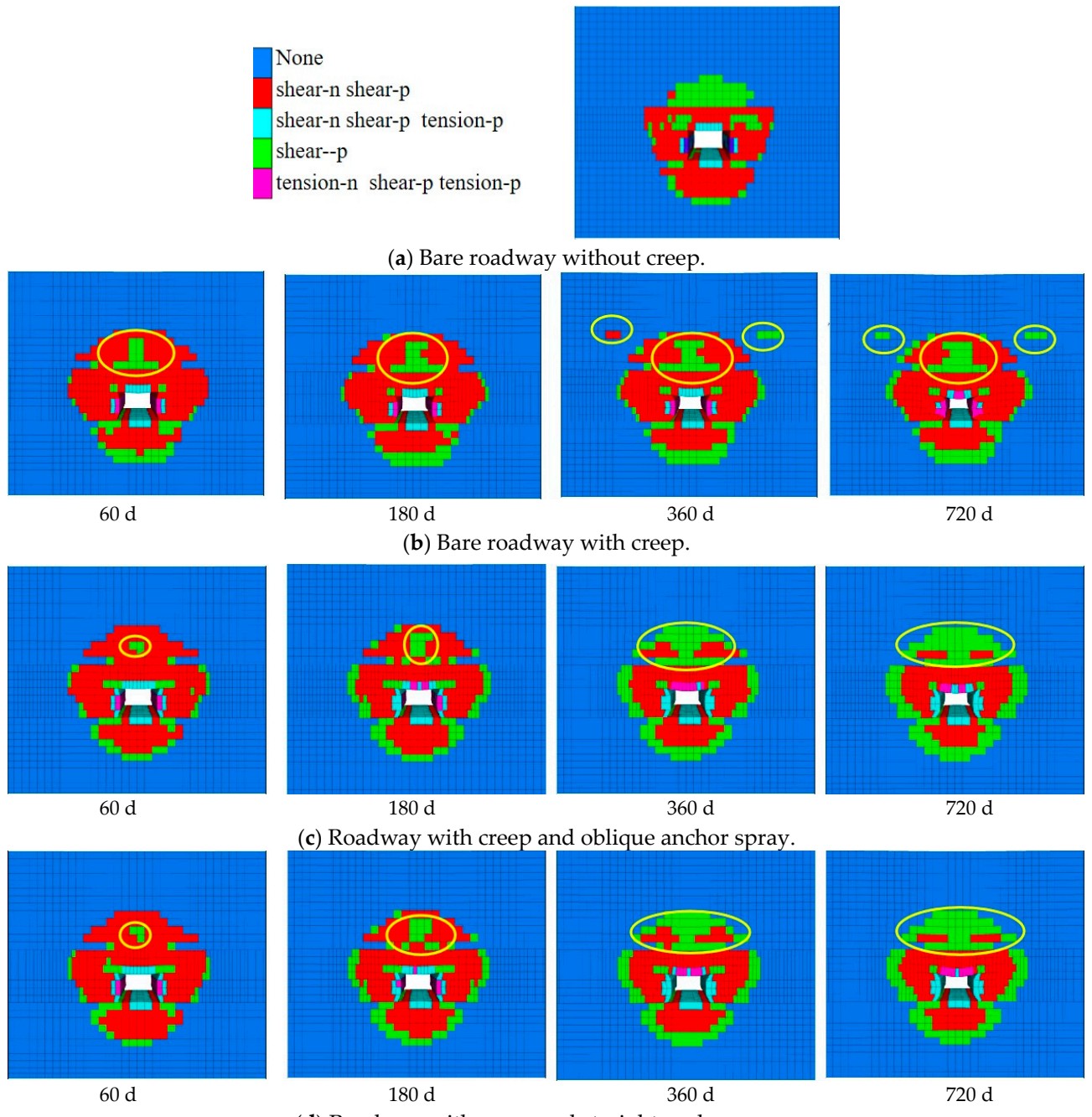

**Figure 11.** Evolution of plastic zone over time.

The plastic zone of the bare roadway with creep participation was larger than that with no creep participation, and the plastic zone boundary of the surrounding rock of creep bare roadway was not constant, while the boundary of the plastic zone gradually expanded with time. The creep damage intensified, and the residual strength and bearing capacity were reduced, which resulted in the roadway's instability under creep. After anchoring and shotcrete, the surrounding rock was closed in time, the residual strength was maintained, and the creep stress threshold of the coal rock increased. The supporting structure was anchored in the elastic zone, forming a stable plastic zone and, thereby, ensuring that the designed cross-section shape of the roadway was maintained.

The volume of plastic zone around the excavation under different numerical simulation schemes is listed in the Table 6.

**Table 6.** The volume of plastic zone.

| Constitutive Model | Roadway Condition | Creep Time/d | Volume of Plastic Zone/m³ |
|---|---|---|---|
| No creep | No support | - | 4774.3 |
| | | 60 | 5761.3 |
| | | 180 | 5874.3 |
| | No support | 360 | 6043.7 |
| | | 720 | 6089.6 |
| | | 60 | 5508.2 |
| Creep | Oblique anchor spray | 180 | 5433.5 |
| | | 360 | 5262.0 |
| | | 720 | 5214.6 |
| | | 60 | 5516.2 |
| | Straight anchor spray | 180 | 5433.5 |
| | | 360 | 5252.5 |
| | | 720 | 5220.2 |

### 4.4. Analysis of Damage Variables

The damage evaluation of the surrounding rock was calculated based on the volume variation of the zones' different mechanical states in the rock mass.

$$D = \frac{V_{damage}}{V_{total}} = \frac{V_{excavtion} + V_{plastic}}{V_{total}} \tag{4}$$

where $V_{damage}$—the volume of the model after excavation, m³; $V_{total}$—the total volume before excavation, m³; $V_{excavation}$—the volume of the excavation, m³; $V_{plastic\ zone}$—the volume of the plastic zone that emerged after excavation, m³; D—damage variable.

The damage variable can be obtained from Equation (4). As shown in Figure 12, the variation in surrounding rock damage with time was as follows: presuming no creep deformation for the bare lane, the damage of the surrounding rock was 0.249.

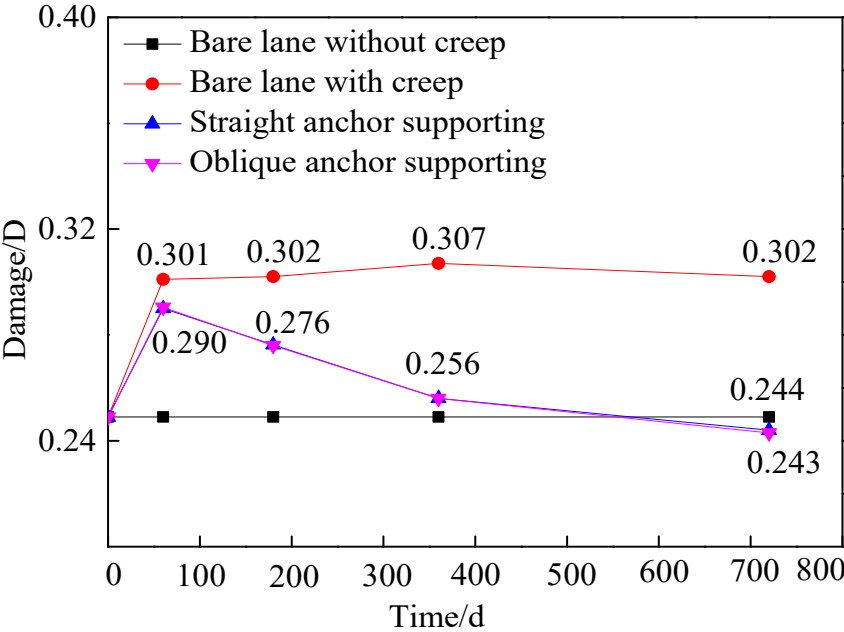

**Figure 12.** Variation in surrounding rock damage with time.

For the bare lane with creep capability, the damage showed a rapidly increasing trend (0~60 d), from 0.249 to 0.301, while a slower increasing trend was observed at 60~360 d, from 0.301 to 0.307; however, the damage later showed a slight attenuation potential (360~720 d), changing from 0.307 to 0.302. Considering bolting and shotcrete, the damage

rapidly increased from 0.249 to 0.290 (0~60 d), while the damage slowly decayed from 0.290 to 0.256 (60~360 d) after bolting and shotcrete. The damage then showed a slight decaying trend (360~720 d), decreasing from 0.256 to 0.243.

The surrounding rock damage rapidly increased in the beginning and then slowly increased to the peak; after that, it slowly decayed. The reasons for this phenomenon were as below: the surrounding rock entered the self-stabilization stage without disturbance as the surrounding rock entered the stable stage, and the longer the self-stabilization time, the lower the damage degree. In the case of supporting, the stability of the surrounding rock further improved, and the degree of damage was reduced.

### 4.5. Analysis of the Displacement Field

As shown in Figure 13, the displacement of the monitoring points changed with time.

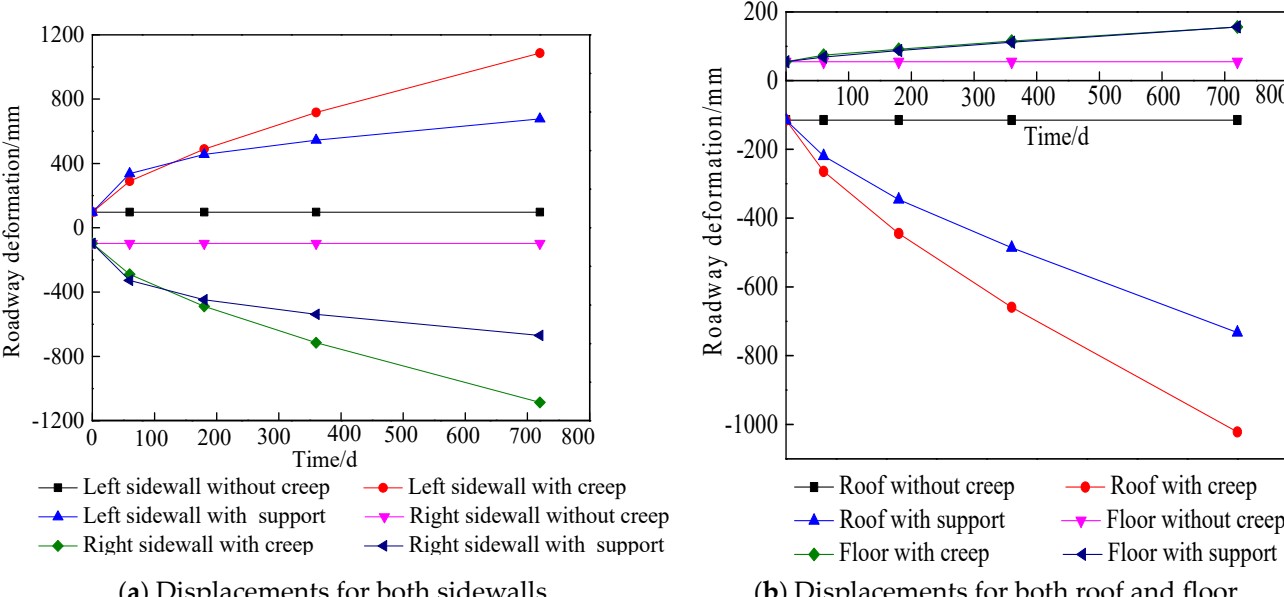

(**a**) Displacements for both sidewalls  (**b**) Displacements for both roof and floor

**Figure 13.** Displacement field change with time.

When the roadway had no creep, the deformation associated with roof subsidence, floor heave, and the two sidewalls' displacement was 115 mm, 55 mm, and 194 mm, respectively.

Under the condition of creep in the rock, the surrounding rock deformation rapidly increased: (i) In the acceleration creep stage (0~60 d), the creep velocity of the roof, floor, and sidewalls was 2.49, 0.32, and 3.22 mm/d, respectively. (ii) In the decline creep stage (60~360 d), the creep velocity of the roof, floor, and sidewalls was, respectively, 1.32, 0.32, and 1.42 mm/d. (iii) In the stable creep stage (360~720 d), the creep velocity of the roof, floor, and sidewalls was 1.01, 0.12, and 1.02 mm/d, respectively. Compared to a lane presumed to have no creep deformation, the order of the roadway deformation reduction was the two sidewalls (1978 mm), the roof (907 mm), and the floor (101 mm).

Under the condition of creep, for the lane with bolting and shotcrete, in the acceleration stage (0~60 d), the creep velocity of the roof, the floor, and the sidewalls of the roadway was 1.75, 0.23, and 4.00 mm/d, respectively. In the decaying stage (60~360 d), the creeping velocity of the roof, the floor, and the sidewalls of the roadway was 0.89, 0.14, and 0.69 mm/d, respectively. In the stable creep stage (360~720 d), the roof, the floor, and the sidewalls' creep velocity was 0.68, 0.10, and 0.37 mm/d, respectively. Compared with the creep-exposed roadway, the reduction in deformation of the roadway ranged, from large to small, as follows: two sidewalls (818 mm) > roof (289 mm) > floor (101 mm).

Compared with no creep, after 720 days of creep in the bare roadway, the amplitude of deformation changed, from large to small, as follows: two sidewalls (1978 mm) > roof (907 mm) > floor (101 mm). The deformation caused by the time effect accounted for 45–88%

of the total deformation, and even after support, it still accounted for a large proportion. This may be due to the creep stress threshold of the floor rock being higher than the creep stress value of the coal. In addition, the roof deforms under the effect of gravity, and the sidewalls deform to the roadway under the effect of horizontal stress. Compared to the creeping bare lane, the amplitude reduction order of the roadway deformation after the anchor spray in different parts of the roadway was as follows: sidewalls (818 mm), roof (290 mm), floor (101 mm). This may be because the support density of the sidewalls is higher than that of the roof after the anchor spray, and it is much higher than that of the floor.

In the acceleration stage, the increase in the plastic zone of the two sidewalls is due to the disturbance when the bolts are installed. With increasing time, the surrounding rock and the bolt fully play their role and increase the creep stress threshold, so the creep velocity is limited.

For the supported roadway, after stabilization, the reduction in the cross-section was not more than 20.4%, which meets the requirements.

## 5. Engineering Applications

### 5.1. Measuring Point Arrangement

Construction was carried out according to the reinforcement support scheme, and the "cross measurement method" was used to monitor the displacement of the 31101 transport slot section, as shown in Figure 14. The shape of the roadway is rectangular, so the measurement points of the roadway were arranged at the midpoints A, B, C, and D to monitor the displacement of the left sidewall, the right sidewall, the roof, and the floor of the roadway. The observer held a laser rangefinder to measure and record the change in the distance from the surrounding rock of the roadway to the center at point O. The surface displacement of the roadway was observed once every two days and the relevant data were recorded for analysis.

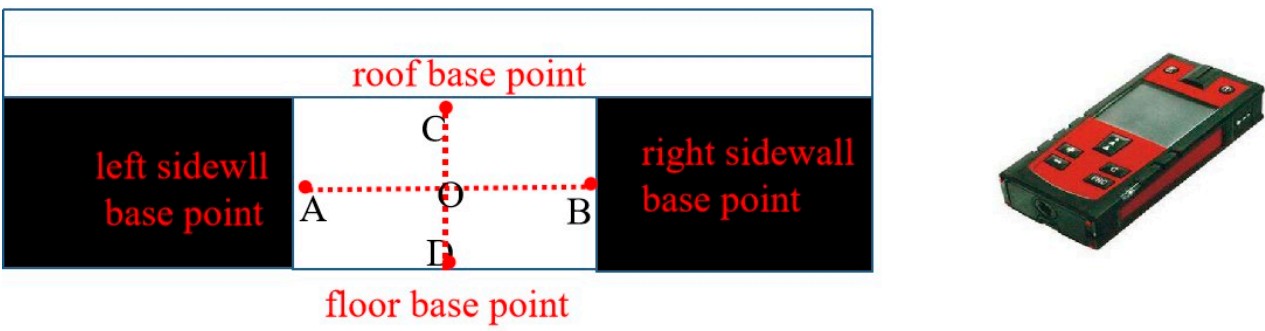

(**a**) Monitoring point arrangement.      (**b**) Laser rangefinder.

**Figure 14.** Monitoring point arrangement and measure instrument.

### 5.2. Model Validation by Engineering Practice

Compared with the displacement of the measured point for the undisturbed roadway with support, it can be seen that the simulation results have good consistency with the measured data. It can be seen that the Cvisc in FLAC simulates flow reliably in a deep high-stress environment. In this paper, an elasto-viscoplastic Cvisc creep constitutive model was firstly confirmed, and then, the model was applied into a FLAC5.0 numerical environment to carry out a simulation base on the creep case of surrounding rock at the 31101 transportation roadway. Finally, the proposed model was well-validated by real engineering practice. The comparisons between field support monitoring data and simulation are shown in Figure 15.

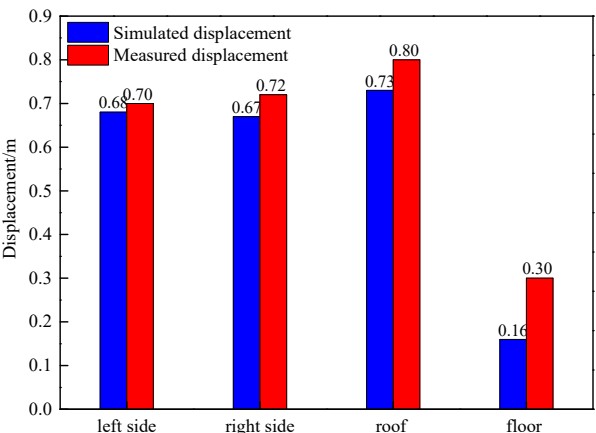

**Figure 15.** Comparison between measured data and simulated results.

## 6. Conclusions

1.   The long-term strength of coal is lower than that of sandy mudstone. The long-term strength of the sandy mudstone and coal is 39.95 MPa and 18.65 MPa, respectively. The creep deformation of coal is more obvious than that of sandy mudstone.

2.   Under the high-stress environment in deep coal mines, creep has a significant influence on the deformation of coal and rock. Under bare roadway conditions, the creep composed 45~88% of the deformation, and the damage caused was 17.5%.

3.   After roadway excavation, the creep of the surrounding rock can be divided into three stages: the acceleration stage (0~60 d), the decaying stage (60~360 d), and the stable stage (360~720 d). In the acceleration stage, the displacement and damage of the surrounding rock increase rapidly; in the decaying stage, the displacement of surrounding rock and the damage increase slowly; in the stable stage, the displacement slowly increases and the damage decreases.

4.   Bolting and shotcrete can effectively suppress the displacement and damage of the surrounding rock. Bolting and shotcrete closed the surrounding rock and maintained the residual strength of the surrounding rock. The support and the surrounding rock together acted as the load-bearing structure; this increased the stress threshold of the surrounding rock rheology and increased the stability of the roadway. In terms of restraining the creep displacement of the surrounding rock, there was no difference between the straight anchor and the oblique anchor. In terms of restraining the shear plastic zone, the oblique anchor had an advantage, and the straight had an advantage in suppressing the tensile plastic zone.

5.   Enhancing the density of the anchor rod support strengthens the surrounding rock in weak areas of the roadway. Strengthening the support provides greater lateral confining pressure to the surrounding rock, effectively improving the long-term strength of the creep zone of the surrounding rock.

6.   The research results reveal the creep characteristics of soft rock under deep high-stress conditions, providing a reference for the study of the long-term stability of coal and rock masses under high-stress conditions. At the same time, it also reminds mining workers to pay attention to the impact of time on mining. (Table 7).

**Table 7.** Main parameters and support density of anchor.

| Name | Preload (kN) | Breaking Load (kN) | Sidewall Support Density (m²/piece) | Roof Support Density (m²/piece) | Floor Support Density (m²/piece) |
|---|---|---|---|---|---|
| Anchor | 60 | 209.4 | 2.5 | 2.14 | 0.71 |

**Author Contributions:** Validation, D.Y.; Writing—original draft, Z.Y.; Writing—review & editing, C.Z. All authors have read and agreed to the published version of the manuscript.

**Funding:** This research was supported by the National Natural Science Foundation of China (No. 52104155) and the Fundamental Research Program of Shanxi Province (Grant No. 20210302124355).

**Informed Consent Statement:** Informed consent was obtained from all subjects involved in this syudy.

**Data Availability Statement:** Not applicable.

**Acknowledgments:** Beijing Computer Center is gratefully acknowledged for providing testing equipment.

**Conflicts of Interest:** The authors declare that they have no conflict of interest.

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
