# Peer review of "Investigation of the Time-Dependent Stability of a Coal Roadway under the Deep High-Stress Condition Based on the Cvisc Creep Model"

_sustainability, doi:10.3390/su151712673_

Round 1
Reviewer 1 Report
The article contains interesting numerical tests on the possible range of rock destruction zones around the excavation depending on the type of mining support and the assumed creep model. The obtained test results were compared with the displacement values in real conditions, for which similar values were received, which may confirm the proposed geomechanical model. Below are some comments and suggestions:
1. In the introduction, it should be mentioned that in the conditions of the rock mass exposed to significant deformations of the rock layers, a special role is played by the mining support, which should be characterized not only by high strength, but above all by flexibility adapted to the moving rocks (doi: 10.3390/en15103774 ; doi: 10.3390/ en15072574);
2. In the subsection 2.1, it should be written how the post-mining space was liquidated (filling or caving). This is quite important information due to the distribution of deformations around mining excavations;
3. In the subsection 2.2, information on fixing rock bolt supports should be added: along the entire length or in sections;
4. For the data shown in Figure 5, it should be added information about the loading rate or strain in the text;
5. In the subsection 4.2, the parameters of the rock bolt support adopted in numerical modeling should be added;
6. In the subsection 4.3.3 for Figure 11, it would be useful to list the results in the table of the extent of the damage zone around the excavation - this would significantly improve the significance of the results obtained;
7. In the analysis of the results regarding displacements (subsection 4.5), it is necessary to indicate what are the limit values in the analyzed excavation - that is, by what percentage can the excavation's cross-section be reduced so that it retains its functionality;
8. In the fifth chapter, it should be written how the displacement measurements were performed: manually or automatically. In addition, please state whether cracking of the floor was also noticed in the mining conditions - this may explain the almost twice higher value compared to the numerical model;
9. In the conclusions, one statement should be added regarding the recommendations for the mine in the field of mining excavation support.
Reviewer 2 Report
The authors present an interesting and important study on stability of coal roadway under high stress with numerical analysis which is emerging for research. The manuscript title “Investigation of the Time-Dependent Stability of Coal Roadway under the Deep High-Stress Condition Based on the Cvisc Creep Model”. Explains about simulation approach using numerical analysis by Cvisc Creep Model with FLAC3D.Overall, the study is interesting.
Undoubtedly, the study is intriguing and valuable to the scientific community. However, there remain some issues and uncertainties that should be addressed before the manuscript can be considered for acceptance. In the following paragraphs, I will raise specific questions and provide constructive feedback to aid the authors in refining their work.
1. What is creep, and why is it considered a fundamental property in some types of rock for roadway stability during mining operations? Any references?
2. Which model was adopted and introduced in FLAC3D to investigate the influence of creep on roadway stability, and why was this particular model chosen?
3. What were the long-term strength values obtained from laboratory experiments for 3-1 coal and sandy mudstone, respectively?
4. After being excavated for 720 days, how did the plastic zone, deformation, and damage of roadway surrounding rock behave, and how did they compare to the initial state?
5. What are the four models used to describe the plastic zone failure, and what are their corresponding volumes?
6. 10. In summary, what were the key findings of the study regarding the influence of creep on roadway stability, and how can the Cvisc creep model contribute to future mining operations?
7. How does the long-term strength of coal compare to sandy mudstone, and what are the specific values obtained from the laboratory experiments?
8. In the high-stress environment of deep coal mines, what percentage of deformation is attributed to creep under bare roadway conditions, and what is the corresponding damage percentage?
9. How do the results of this study align with or differ from previous research on creep behavior in similar geological conditions? Kindly provide some references.
In the conclusions, authors should explain the importance of the study, its future scope, and its contribution to the national and international research community.
Overall, the English used in the review is clear and coherent, effectively conveying the subject of the research paper.
However, there are a few instances where sentences could be further refined to enhance readability and flow.
With the suggested improvements in sentence structures for better readability, the review becomes even more polished and impactful.
Round 2
Reviewer 2 Report
Dear Authors
Thank you for your revised paper. I have carefully read the revised paper and found it has been substantially improved by responding to and clarifying the queries raised by the reviewers.
i have only one minor revision, add this point answer to your manuscript.
In the conclusions, authors should explain the importance of the study, its future scope, and its contribution to the national and international research community.
Overall good
